# Corrosion Resistance of Nickel-Aluminum Sinters Produced by High-Pressure HPHT/SPS Method

**DOI:** 10.3390/ma16051907

**Published:** 2023-02-25

**Authors:** Paweł Hyjek, Michał Stępień, Remigiusz Kowalik, Iwona Sulima

**Affiliations:** 1Institute of Technology, Pedagogical University of Krakow, Podchorazych 2 St., 30-084 Krakow, Poland; 2Faculty of Non-Ferrous Metals, AGH University of Science and Technology, Mickiewicza 30 Av., 30-059 Krakow, Poland

**Keywords:** corrosion, HP SPS, sintering, innovative high-pressure processes, Ni-Al

## Abstract

As part of extensive research on the properties of nickel-aluminum alloys, corrosion tests of sintered materials produced by the innovative HPHT/SPS (high pressure, high temperature/spark plasma sintering) method were performed in 0.1 molar H_2_SO_4_ acid. The hybrid, unique device used for this purpose (one of only two such devices operating in the world) is equipped with a Bridgman chamber, which allows heating with high-frequency pulsed current and sintering of powders under high pressure in the range of 4–8 GPa and at temperatures up to 2400 °C. Using this device for the production of materials contributes to the generation of new phases not obtainable by classical methods. In this article, the first test results obtained for the nickel-aluminum alloys never before produced by this method are discussed. Alloys containing 25 at.% Al, 37 at.% Al and 50 at.% Al were produced. The alloys were obtained by the combined effect of the pressure of 7 GPa and the temperature of 1200 °C generated by the pulsed current. The time of the sintering process was 60 s. The electrochemical tests, such as OCP (open circuit potential), polarization tests and EIS (electrochemical impedance spectroscopy), were carried out for the newly produced sinters and the results were compared with the reference materials, i.e., nickel and aluminum. The corrosion tests showed good corrosion resistance of the produced sinters, with corrosion rates of 0.091, 0.073 and 0.127 mm per year, respectively. It leaves no doubt that the good resistance of materials synthesized by powder metallurgy is due to the proper selection of the manufacturing process parameters, ensuring a high degree of material consolidation. This was further confirmed by the examinations of microstructure (optical microscopy and scanning electron microscopy) and the results of density tests (hydrostatic method). It has been shown that the obtained sinters were characterized by a compact, homogeneous and pore-free structure, though at the same time differentiated and multi-phase, while the densities of individual alloys reached a level close to the theoretical values. The Vickers hardness of the alloys was 334, 399 and 486 HV10, respectively.

## 1. Introduction

For many years, Ni-Al alloys have been perceived as interesting construction materials to be used for machine and device components, mainly in the manufacturing [1,2], energy [3,4], automotive [5,6] and aerospace [7,8] industries. The popularity of these materials is due to their relatively low density and high strength [1,2,9], especially at high temperatures [7,10,11]. An essential advantage of Ni-Al alloys that promotes their wide application is their good corrosion resistance [1,9,12]. The inherent brittleness at ambient temperature can be counteracted by the use of some alloy additives, to mention as examples boron [13,14], boron and zirconium [15], titanium [16] and iron, which is an impurity generated by the steel grinding media and is the source of serious disorders in the microstructure of NiAl [17], copper [18], or carbon [19], and also by the use of appropriate manufacturing methods. Special attention in the manufacture of Ni-Al alloys deserves the technology of powder metallurgy as an important tool that allows abating the adverse effect of brittleness [2] through the use of various manufacturing methods [20,21], microstructure optimization [22,23] or ultra-fine grain refinement [24]. Numerous variations of the sintering process have been used to meet this goal, starting with free sintering [25,26], through the techniques that use both high sintering temperatures (HP—hot pressing, HIP—hot isostatic pressing) [22,27,28] and high sintering pressure (HPHT—high pressure high temperature) [29,30], and ending in the methods where thermal effects are obtained by the use of direct current, alternating current or pulsed current (SPS—spark plasma sintering) [31,32,33]. Currently, scientists are particularly interested in the SPS method. After numerous modifications [34], this method is considered a relatively modern production tool that contributes to the manufacture of materials characterized by low porosity and is achieved at lower parameters of the sintering process (mainly temperature) than the parameters used in other more traditional methods [35]. High-pressure spark plasma sintering (HP (HT)/SPS) is an innovative variant of this method. Using this and other new methods based on SPS, it is possible to synthesize virtually all materials, including those with metastable or intermetallic phases and composites, such as HfB_2_/SiC and HfB_2_/HfC/SiC [36], MoSi_2_@ZrO_2_ [37], NiAl/Ni_3_Al/TiB_2_ [38], TiB2-B4C with hBN [39] and Ti-6Ni-xTiCN [40]. The HP/SPS method belongs to the new generation of SPS processes and has already been successfully used for the sintering of polycrystalline diamond [41], nanocrystalline γ-Al_2_O_3_ [42], β-SiC [43], ZrC-based composites [44] and Ti–Al–Si alloys [45].

It has been shown that the use of high pressure [46,47] and high temperature in the sintering process produces microstructures that cannot be obtained by traditional sintering methods [48,49]. The application of high pressure (up to 12 GPa [50]) or the use of special chambers ensuring the required distribution of pressure and temperature, combined with different techniques of heat supply and discharge, contribute to the formation of different microstructures and, consequently, different properties of otherwise identical alloys. It has to be remembered that in the manufacturing process of Ni-Al alloys, at a certain temperature, a sudden explosive exothermic transformation (heat release) occurs, accompanied by a rapid increase in temperature. As a result of this phenomenon, the final composition of the manufactured alloy is difficult to control, but this type of reaction of synthesis is often used to reduce production costs and shorten the duration of the entire process [51,52], reducing also in this way the risk of grain growth [53]. Additionally, it can be expected that, by extending the range of plastic deformation, the high pressure used in the process will confer on the initially brittle materials’ (intermetallic compounds) better mechanical properties after the process.

As part of the conducted research, Ni-Al intermetallics with a composition close to 25 at.% Al, 37 at.% Al and 50 at.% Al were produced. Sintering with the simultaneous effect of pressure and temperature was carried out in a globally unique HPHT/SPS hybrid device equipped with a Bridgman chamber. The principle of operation of this device has recently been described by Guignard et al. [41] in a study where the applied method was called ultra-high-pressure (UHP)-SPS.

When planning the research, it was assumed that the use of high pressure (7 GPa) in the manufacture of the above-mentioned alloys, the use of appropriate chambers ensuring the required distribution of pressure and temperature, the use of pulsed current as a means of heat supply and, finally, the violent exothermic reaction (temperature increase) taking place at the nickel/aluminum interface at a temperature of approx. 575 °C [33] would contribute to the manufacture of material characterized by a high degree of consolidation (almost pore-free). This, in turn, as indicated by Went et al. who claimed that “*Microstructure has an inseparable relationship with corrosion resistance*” [54], should contribute to better corrosion resistance of the alloys. Osorio et al. [55] also noticed this relationship in their research. They linked the microstructure to the electrochemical corrosion resistance of Ni-Al alloys that solidified directionally and showed that the dendritic system and the distribution of NiAl_3_ intermetallic particles are of great importance for both the pitting potential and the overall corrosion resistance of Al-Ni alloys. Compared to the fine-grained dendritic microstructure after casting, the coarse-grained dendritic microstructure promotes overall corrosion/oxidation resistance, though Zhang et al. [56] suggested that the more homogeneous and fine α-Al/Al_3_Ni microstructure plays an important role in the improvement of corrosion resistance of the Al-5.4 wt.% Ni alloy. 

In [57,58], a comparison was made between the corrosion resistance of Ni-Al coatings and composite Ni-Al-Al_2_O_3_ coatings in an environment of 0.01 M H_2_SO_4_ and 3.5% NaCl. The evaluation based on polarization and impedance spectroscopy measurements showed that the examined coatings were more resistant to the 3.5% NaCl environment than to the 0.01 M H_2_SO_4_ environment. As a consequence of this report, it was decided to base the assessment of the corrosion resistance of the alloys described in this article on tests carried out in a more aggressive environment of H_2_SO_4_ acid.

## 2. Materials and Methods

### 2.1. Selection of Starting Powders

Commercial carbonyl nickel powder (Vale prod., Clydach, UK) (Figure 1a) and gas-atomized aluminum powder (Benda-Lutz prod., Skawina, Poland) (Figure 1b) with a purity of 99.8% and particle sizes of 3–7 µm and 32 µm, respectively, were used to produce the sintered material. It is easy to observe that the morphology of the nickel powder is close to a spherical shape and that the particles are characterized by high repeatability of both size and shape. On the other hand, aluminum powder particles are more diverse in both size and shape, though they tend to occur in the form of elongated rods. To hinder oxidation, the aluminum powder particles were larger in size than the nickel powder particles. This type of relationship in the size of Ni and Al particles was indicated by Philpot et al. [59] and Biswas et al. [60], who studied the temperature of the explosive reaction and its effect on the formation of a single-phase microstructure.

### 2.2. Preparation of Powder Mixtures

To produce the required alloys, designated according to the aluminum content expressed in atomic percent as NiAl25, NiAl37 and NiAl50, nickel and aluminum powders were weighed in appropriate proportions using a balance with an accuracy of 0.0001 g, model AS 220/C/2, created by Radwag (Radom, Poland). To obtain reasonably homogeneous mixtures, mixing was carried out for 24 h in a Turbula WAB Type T2F mixer (Willy A. Bachofen AG, Muttenz, Switzerland) using AISI52100 steel balls in a 2:1 weight ratio in relation to the powders.

### 2.3. Fabrication of Test Materials

#### 2.3.1. Description of the High-Pressure HPHT/SPS Sintering Process

Individual mixtures in appropriate amounts (determined by their specific volume) were placed in a special container (Figure 2). A photo of the container (before final assembly of the outer ceramic part 1 and inner ceramic part 2) is shown in Figure 2a, while Figure 2b shows a schematic diagram of the container. According to Klimczyk [61], proper design of containers contributes to the uniform distribution of compressive stresses as a result of plastic deformation of the gaskets (1–3) and to uniform heat distribution (graphite resistance heater 4), thus allowing for the imitation of quasihydrostatic process conditions in the sintered material (6).

Gaskets are composed of special metamorphic rocks (catlinite, pyrophyllite and lithographic stone) [62]. The volume of the toroidal chamber (0.3–1 cm^3^) enables generating both high pressure (up to 12 GPa) and high temperature (up to approx. 2500 °C). This system is most often used for the production of synthetic diamonds [63], PCD (composites based on polycrystalline diamonds) [64,65], cBN (regular boron nitride) and PcBN (polycrystalline cubic boron nitride) [61]. As a result of the simultaneous effects of pressure and temperature, the sintering process is much faster than in the case of free sintering. The typical duration of the HPHT sintering process is approximately 0.5–2 min, while free sintering requires longer times, from several dozen minutes to several hours [50]. The short duration of the process contributes to the reduction in grain growth, which is important when the sintering of nanopowders is carried out. Materials obtained by the high-pressure method are characterized by a nearly 100% densification degree, isotropic properties and, in some cases, because of the different thermodynamic conditions of the production process, a phase composition completely different from the phase composition of the same material subjected to free sintering [30,46,50].

#### 2.3.2. Process Parameters

The test material was produced by high-pressure spark plasma sintering. For this purpose, a modified HPHT/SPS apparatus (Łukasiewicz Research Network—Krakow Institute of Technology, Krakow, Poland) was used. It included a hydraulic press equipped with a Bridgman-type anvil and a pulsed DC generator. The containers were placed between the anvils (Figure 2c). The Bridgman-type anvil has a toroidal shape, which produces a quasi-isostatic pressure applied to the material as a result of plastic deformation of the gasket with sinter. Heating is generated by a 1 kHz pulsed current that flows directly through the graphite heater (4) in the gasket and the conductive sintered material. Compared to conventional sintering methods, this method of heating (pulsed current) offers some significant advantages [66,67], to mention only the possibility of using lower sintering temperatures and generating new phases as a consequence of the very high heating/cooling rates and surface activation of powders by the plasma cleaning method [34,68,69].

The following test parameters were applied: pressure P = 7 GPa, temperature T = 1200 °C, heating—5 s, proper sintering—60 s and cooling—5 s. As in the case of the research on the manufacture of Ti-Al-Si alloys [45], the temperature was set by proper calibration for a pulse length of 40 ms and a pulse interval of 20 ms. It changed with the changing degree of pulse duty. The duty cycle is a programmable parameter of the HPHT/SPS device that describes the duty cycle of a programmed 40 ms 1kHz pulsed DC pulse with 1ms initial pulses. Temperature calibration is necessary since there is no possibility to measure temperature directly during the sintering process. 

### 2.4. Research Methods

The test materials, in the form of disks with a diameter of 15 mm and a height of 5 mm, were machined (grinded) to size and then cleaned and degreased for testing. The initial visual assessment and the density measurements enabled estimating and determining the quality of workmanship and the degree of sintering, including the level of porosity. Density tests were carried out by the hydrostatic method, using the above-mentioned balance equipped with an appropriate adapter for the determination of the density of solids. Hardness measurements were carried out under a load of 98.1 N using a NEXUS 4000 hardness tester (INNOVATEST EUROPE BV, Maastricht, The Netherlands). The microstructure of the produced sinters was examined with an Olympus GX51 (Tokyo, Japan) optical microscope and a JEOL JSM 6610LV (Tokyo, Japan) scanning electron microscope with an EDS-Oxford analyzer.

The electrochemical characterization was conducted in a 0.1 M H_2_SO_4_ solution using an AUTOLAB PGSTAST 302n potentiostat with an FRA2 module (AUTOLAB, Utrecht, The Netherlands). A conventional three-electrode electrochemical cell with an Ag/AgCl reference electrode and a platinum sheet counter electrode was applied. The specimens were mounted in epoxy resin with an electrical connection on one side. An 8-mm PTFE gasket was used to delimit the working area of each sample (area = 0.5026 cm^2^). Before each experiment, the upper surface was ground with an 800-grit abrasive paper and cleaned in an ultrasonic bath with water. To compare the corrosion properties, commercially pure nickel (Alloy 200, Bibus Metal, Dąbrowa, Polska) and aluminum (Alloy 1050, BimoTech, Wrocław, Polska) were also tested.

Before polarization tests, each sample was conditioned for 120 s in a 0.1-M H_2_SO_4_ solution to get an open circuit potential (OCP). The potentiodynamic polarization scan was performed from −0.25 V vs. OCP to a potential where the current density reached 10 mA/cm^2^ at a rate of 1 mV/s. The Tafel slopes were calculated from the active regions of the corresponding anodic and cathodic curves. The experiments were repeated at least three times to check the reproducibility of the results.

Electrochemical impedance spectroscopy (EIS) and OCP measurements were conducted concurrently. The EIS spectra were obtained after immersing for 5 min, 1 h, 2 h, 4 h, 8 h, 16 h and 24 h with a frequency range from 100 kHz to 1 Hz and a sinusoidal voltage of 10 mV amplitude. Between EIS measurements, the cell was switched off and OCP measurements were continued. All measurements were carried out at 25 ± 2 °C in a naturally aerated solution. The obtained EIS spectra were fitted to the chosen equivalent circuit model with Z-view software.

## 3. Results and Discussion

### 3.1. Microstructural Analysis

Microstructural examinations carried out by optical microscopy and SEM allowed for determining the degree of material sintering (consolidation). No pores or discontinuities were found on the surface of the produced materials. The examinations revealed a homogeneous, multiphase structure of the manufactured alloys. However, this was not consistent with the phase structure under equilibrium conditions in the Ni-Al phase diagram, which shows that a single Ni_3_Al phase can be obtained with n(Ni):n(Al) = 3:1 [70], and likewise a single NiAl phase can be obtained with n(Ni):n(Al) = 1:1 [54,71]. Similar microstructures were obtained by Cui et al. [72].

Examples of images taken by optical microscopy and SEM are shown in Figure 3a (NiAl25), Figure 4a (NiAl37), Figure 5a (NiAl50) and in Figure 3b, Figure 4b and Figure 5b, respectively. The EDS analysis (area, point, line and map) enabled the identification of the examined areas, which were characterized by large variations in the aluminum content. The probable cause of these variations is the use of high pressure during sintering. Combined with an explosive reaction (mentioned by other authors in [52,73,74,75], where various mechanisms of the phase formation in an Ni-Al system were discussed), the effect of high pressure may, depending on the occurrence of some specific parameters, contribute to the formation of a multiphase microstructure. These processes were also described by A. Biswas et al. [60,70], who emphasized the fact that the size of nickel particles and the rate of heating are to be included among the most essential solid-state processing parameters that play a major role in the thermal explosion of NiAl.

To confirm the impact of high pressure on the microstructure obtained in the HPHT/SPS process, the results of parallel studies were analyzed. Using the same mixtures, sintering was performed by the traditional FAST/SPS method (FCT HP5 device), where the explosive reaction also occurs during sintering (RSPS-reactive SPS). The following test parameters were applied: a pressure of 48 MPa and a temperature of 1200 °C. As a result of this process, materials with a single-phase microstructure were obtained (Ni_3_Al for the Ni75Al25 alloy and NiAl for the Ni50Al50 alloy). The alloy with 37% Al had a two-phase Ni_3_Al/NiAl microstructure. Yet another type of microstructure was revealed by the research described in [76]. Numerous studies and tests show that whenever high sintering parameters (pressure and temperature) are used in the sintering process and, additionally, an explosive reaction occurs, it is difficult to control the composition and microstructure of the obtained material, since too many factors are involved in the process [52]. This was pointed out by Thompson et al. [77], who described the effect of alloy addition on the microstructure evolution in Ni-Al alloys. The same was also observed by the author in his own research [33], where the introduced ceramic particles were found to have an effect on the composite microstructure formed as a result of combustion synthesis (CS). Ozdemir [78] obtained an almost identical microstructure of the two-phase NiAl-Ni_3_Al alloy with a molar ratio of Ni and Al equal to 13:7.

A general conclusion from the research findings is that the microstructure of the NiAl25 alloy is fairly homogeneous (Figure 3a) and consists of a minimum of three phases (Figure 3b), including, as indicated by the EDS analysis, an aluminum-rich phase, an intermediate phase and a nickel-rich phase with unreacted nickel.

The microstructure of the NiAl37 alloy is homogeneous (Figure 4a) and similar to a two-phase microstructure (Figure 4b), where phases rich and poor in aluminum exist together. As shown by previous [33] and parallel studies (FAST/SPS), these are most likely the NiAl and Ni_3_Al phases, respectively.

The microstructure of the NiAl50 alloy is also homogeneous (Figure 5a) and is similar to a single-phase microstructure (Figure 5b). According to the equilibrium diagram, this is the NiAl phase, although occasionally areas rich in nickel (with traces of aluminum) are also present.

### 3.2. Density and Hardness Measurements

The results of the density measurements carried out on the obtained sinters are presented in Table 1.

The results confirm that the conclusions drawn from the microstructural examinations of the test materials, indicating a high degree of consolidation obtained in the produced sinters (the density being close to the theoretical values), are correct. The factors responsible are the high parameters of the sintering process, i.e., the temperature of 1200 °C and pressure of 7 GPa, as indicated by [29] and the authors’ own research [79]. Comparing the results of the density measurements with the results given in [80], where the test material was created by casting, free sintering [81], mechanical alloying (MA), HP [82] and SPS [83], it has been found that the HPHT/SPS device used for the manufacture of Ni-Al alloys produces materials with a much higher degree of consolidation. Additionally, despite the fact that the obtained materials had different microstructures, no differences were observed in the mechanical properties, as indicated by the results of the HV10 hardness measurements. The largest scatter of results (and thus the heterogeneity of properties) was observed in the two-phase NiAl37 alloy, but even in this case, the values were relatively small (1.4%). Additionally, the results of hardness measurements gave values different from the values stated by, e.g., Cymerman et al. [76]. For Ni_3_Al and NiAl, they obtained 305 HV10 and 290 HV10, respectively, and both values were achieved at a sintering temperature of 1000 °C. At higher temperatures, the results were even lower. As indicated by the examinations of microstructure, the high values of hardness obtained in the research may be due to the synergy effect caused by the occurrence/precipitation of individual phases in some specific areas and/or the strengthening effect associated with high sintering pressure. However, since the highest hardness values were obtained for the last single-phase alloy (NiAl50), it seems that the latter effect associated with the use of high pressure in the manufacturing process is of major significance here.

### 3.3. Analysis of Corrosion Behavior

To determine the corrosion resistance of Ni-Al alloys produced by the HPHT/SPS method, electrochemical tests were performed in a 0.1-M H_2_SO_4_ solution. The change in the open circuit potential (OCP) during the immersion of samples in the solution for 24 h is shown in Figure 6. For comparative purposes, the tests were also carried out on pure nickel and aluminum. The OCP values recorded for a selected group of materials are very similar and their change over time is insignificant. A small increase in potential is observed at the beginning of the measurement, and it is caused by an oxide layer formed on the surface.

The value of the potential stabilizes after 8 h. Comparing the obtained values, it has been noted that they are very close to the values recorded for pure nickel. The aluminum content in the alloy reduces, but the OCP value reduces only slightly. The comparison of NiAl25, NiAl37 and NiAl50 samples shows that the last sample has a lower corrosion potential, probably due to the higher aluminum content. The high OCP values recorded for all materials may result from the dissolution of a less noble metal (Al) and the following enrichment of the surface layer in a more noble metal (Ni) [84]. The surface of the nickel-rich material is then passivated in a solution of sulfuric acid and a tight oxide layer is formed. The high stability of the recorded OCP indicates the high durability of the passive layer. The slightly lower potential of sintered samples compared to pure nickel may result from both sample composition and microstructure. Despite the density values being close to the theoretical values, the microstructure of sintered materials is different from the microstructure of materials obtained by conventional methods. The material is more fragmented, which can also trigger the appearance of additional effects. An example may be found in the research where it has been demonstrated that sintered materials are characterized by a thicker passive layer formed on their surface [85,86].

To determine the mechanism and kinetics of the corrosion process taking place in the tested materials, potentiodynamic measurements were carried out and polarization curves were plotted (Figure 7). All samples were passivated in sulfuric acid solutions. However, with the aluminum content increasing in the samples, the shape of the polarization curves changed in the range of passive layer formation. The most significant changes were observed in the range of potentials responsible for forming a passive layer.

In the case of the NiAl25 sample (Figure 7a), the shape of the polarization curve was the same as for pure nickel (Figure 7d). The passive layer began to form after exceeding the corrosion potential. The presence of two peaks on the polarization curve indicates a complex mechanism of the material passivation process resulting from the formation of several corrosion products. Nickel oxides NiO and Ni_3_O_4_ [87,88], as well as hydroxides Ni(OH)_2_ [89,90], can appear on the surface of nickel during corrosion in sulfuric acid solution. The content of 25 at.% Al does not significantly alter the mechanism of passive layer formation, but its importance is visible in the area of transpassivation. The current value recorded in the range of potentials where nickel transpassivation takes place is lower than in the case of pure nickel. The presence of aluminum in the alloy inhibits further oxidation of nickel in the transpassive region.

Aluminum content increasing in the samples visibly affects the corrosion mechanism. In the samples containing 37 at.% and 50 at.% Al, oscillations occur in the range of active dissolution and this effect may be due to the oxidation of aluminum (sample NiAl37—Figure 7b, sample NiAl50—Figure 7c). For comparison, pure aluminum is oxidized in sulfuric acid solution, and the characteristic active state/passive state transition in the form of a peak (Figure 7e) does not appear on the polarization curve, as is the case with pure nickel (Figure 7d). During the polarization of a pure aluminum electrode, the thickness of the oxide layer increases with time [91]. Therefore, in the range of more positive potentials, where passivation of the remaining samples is observed, the value of the recorded current is higher for the electrode composed of pure aluminum. When the aluminum content in the sample is 25 at.%, passivation occurs very quickly, and the oxidation process is similar to that of pure nickel. Aluminum and nickel have different electrochemical potentials. Consequently, corrosion microcells are formed between individual phases. Due to its lower electrochemical potential, aluminum is the first to dissolve under the effect of the applied voltage. The process advances until the surface layer is enriched in nickel and then covered with a tight passive layer [84]. Increasing aluminum content prolongs this process, and its course is signaled by the appearance of oscillations. Despite the aluminum content increasing up to 50 at.%, each of the alloys was transformed into a passive state. Only when the aluminum content of the material was 50 at.% Al, the process of the passive layer formation lasted longer and the passivation current was higher than in other alloys. Additionally, when the aluminum content in the alloy was 37 at.% and 50 at.%, a characteristic effect of the disappearance of the transpassive region occurred. Based on the polarization curves, the corrosion potential and corrosion current values were determined (Figure 8). The corrosion current systematically increased with the aluminum content, and the corrosion potential decreased with the aluminum content. The results indicate that, among the tested alloys, the NiAl25 (25 at.% Al) sample was characterized by the highest corrosion resistance in the sulfuric acid environment. The value of the corrosion current also indicates that the corrosion resistance of the NiAl25 sample was slightly superior to the resistance of pure nickel.

The microstructure of materials obtained by powder metallurgy differs from the microstructure of materials obtained by conventional metallurgical methods, which also impacts the corrosion behavior of the tested alloys [85,92,93,94]. The selected synthesis method of Ni-Al alloys can induce the formation of pores in the material, which may act as a site of crevice corrosion [94]. Therefore, polarization curves were plotted 24 h after immersing the samples in a sulfuric acid solution. In none of the cases, significant differences in the shape of the polarization curves obtained immediately after immersion and plotted after the lapse of 24 h were observed. The shape of the polarization curves suggests a shifting of the process of active oxidation of the tested material towards more positive potentials. Since plotting the polarization curves starting from the potential of −0.25 V vs. OCP, part of the passive layer has already been reduced. When the corrosion potential was exceeded, the passive layer formation started once again, but the first peak on the polarization curves obtained for the alloy samples in the range of active oxidation was missing. This confirms the assumption that the passive layer was not completely reduced.

The corrosion potential is slightly shifted towards more positive values, which is due to the presence of a passive layer formed after 24 h. The values of the corrosion current are much lower than the values determined from the polarization curves obtained immediately after immersion of the samples in the solution. It indicates that the passive layer already existed on the surface of the alloy and very effectively inhibited the alloy dissolution process. Some attention is given due to the fact that the values of the corrosion current are very close to the values obtained for pure nickel, though they are lower than the values obtained for pure aluminum. The results of the corrosion rate (*CR*) calculations based on corrosion current are presented in Table 2.

The calculations were based on Equation (1) according to the ASTM G59-97 Standard.
(1)$$CR=3.27\times 10^{-3}\times \frac{i_{corr}\times EW}{\rho }\;\left[\mathrm{mm}\;\mathrm{per}\;\mathrm{year}\right]$$
where *i_corr_* is the corrosion current in uA/cm^2^, *ρ* is the density in g/cm^3^ and *EW* is the equivalent weight of the examined sample in grams.

The calculated corrosion rates confirm previous conclusions based on polarization curves and corrosion currents. The results were obtained despite significant differences in sample densities (e.g., Ni—8.91 g/cm^3^ and Al 2.70 g/cm^3^). For sintered samples, after 24 h of exposure to the electrolyte, the corrosion rate decreased from 2.5 to 4.5 times compared to the rate after immersion. For the nickel sample after passivation, the corrosion rate decreased approximately five times. In the case of the aluminum sample, taking into account the accuracy of electrochemical methods, the change in the corrosion rate was insignificant. Based on the PN-H-04608:1978 Polish Standard describing the corrosion resistance scale, the corrosion durability of the obtained alloys can be estimated at one to ten years.

The corrosion resistance of the tested alloys was also monitored by EIS. To confirm the high resistance and electrochemical stability of the passive layer formed on the tested sintered materials, measurements were taken at the following time intervals: 5 min, 1 h, 2 h, 4 h, 8 h, 16 h and 24 h. The results of the impedance tests are shown in the form of Nyquist plots (Figure 9). The results are arranged in characteristic flattened semicircles, suggesting charge transfer control, which means uniform corrosion on a homogeneous surface. The discussed case can be described with an equivalent circuit shown in Figure 10, where *R_s_* is the resistance of the electrolyte (including the resistance of wires, etc.), *R_p_* is the polarization resistance and *Z_CPE_* is related to the capacitance of the double layer.

The simple capacitance was replaced with a constant-phase element to allow for the phenomena occurring at the interface and resulting either from the heterogeneity of structure or from the diversity of local chemical composition [95]. The impedance of the constant-phase element is given by Equation (2):
(2)$$Z_{CPE}=\frac{1}{Y_{0}{\left(j\omega \right)}^{n}}$$
where *ω* is the circular frequency, *n* is the exponent showing the degree of surface heterogeneity (the closer *n* is to 1, the more homogeneous the surface is), *Y*_0_ can be directly identified with capacitance when the value of *n* measured in corrosion tests is between 0.9 and 1 [96], and this parameter has been discussed in the present study.

The Bode representation of the impedance data, gathered in Figure 11, shows a single time constant for all the samples (one peak in phase shift curves). Using a single-layer model makes it possible to describe the tested system [97]. Moreover, this means that the electrolyte has reached the barrier layer, no diffusion processes occur and the porosity of the sintered samples is close to zero. The phase shift value tends to increase throughout the whole corrosion test. After 24 h, the phase shift is almost angular (−80°). This shows that the constant phase element behaves similarly to a pure capacitor (*n* = 1, *Rp*→∞, and the phase shift equals −90°). It indicates that the process of passivation is underway. The modulus is much less sensitive and accurate for estimating model parameters. However, a simple increase in the value of |Z| at low frequencies can be equated with an increase in resistance to corrosion processes. The values |Z| for all tested samples showed an increasing trend during the experiment.

Table 3 provides the fitted values of the equivalent circuit parameters for the tested samples. The diameters of the semicircles increasing over time indicate an increase in the polarization resistance *R_p_* and thus in the corrosion resistance of the tested alloys, which may indicate that the passive layer is systematically growing on the alloy surface [98]. This behavior justifies the increase in the value of parameter *n*, especially in the case of reference samples composed of pure metals. It explains the constant OCP potential (Figure 6) and confirms the potentiodynamic results and the calculated values of corrosion current (Figure 8). Ni-Al alloys undergo passivation in the presence of sulfuric acid, and the resulting passive layer is stable and effectively inhibits the corrosion process. Despite the presence of two chemical elements with a fairly large difference in electrochemical potentials, it can be noticed that the passivation mechanism occurs even in the case of samples containing 50 at.% Al. There was no drastic decrease in the corrosion resistance of the sintered material after 24 h of exposure to the electrolyte, and no diffusion effect [99]. Undoubtedly, the reason for the high resistance of Ni-Al alloys synthesized by powder metallurgy is the selection of process parameters that ensure a high degree of material consolidation. Owing to the properly selected parameters of the synthesis process, there is no risk of crevice corrosion in the tested materials, which is very common in materials synthesized by powder metallurgy methods [94].

## 4. Conclusions

Using a modified hybrid HPHT/SPS sintering device, it is possible to manufacture nickel-aluminum alloys from the starting powders with a variable Ni/Al ratio (75/25, 63/37 or 50/50 at.%). The obtained materials are characterized by a diverse, multiphase microstructure, which is due to the effects of high pressure (7 GPa) and pulsed heating (1200 °C) applied in the sintering process. The high pressure and pulsed heating, additionally supported by the exothermic reaction that occurs during sintering, allow for the generation of a material whose microstructure differs from the microstructure determined by the Ni-Al equilibrium system. The test parameters also contribute to manufacturing homogeneous and pore-free materials, which impacts their high overall density (close to the theoretical density), high HV10 hardness and good corrosion resistance. All sintered samples reached the passive state after 24 h of exposure. After 24 h of exposure, the sintered sample of NiAl37 showed the lowest corrosion rate. The obtained value was 0.073 mm per year and was close to the values obtained for pure nickel, i.e., 0.066 mm per year. The densities were 7.38, 6.67 and 5.8 g/cm^3^, respectively, and the Vickers hardness was 334, 399 and 486 HV10, respectively.

Research on the HPHT/SPS production of nickel-aluminum alloys will continue. In particular, the effect of sintering parameters (mainly high pressure) on the formation of specific phases and their quantitative ratio in the alloy will be determined.

## Figures and Tables

**Figure 1 materials-16-01907-f001:**
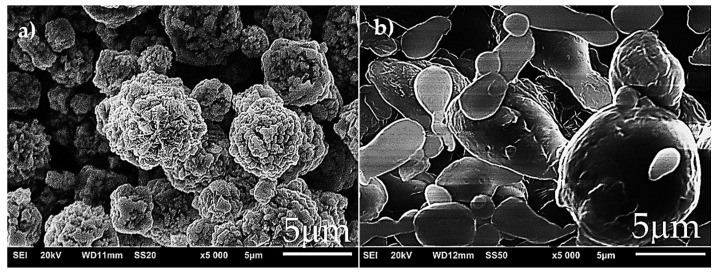
SEM micrographs of (**a**) nickel and (**b**) aluminum powders.

**Figure 2 materials-16-01907-f002:**
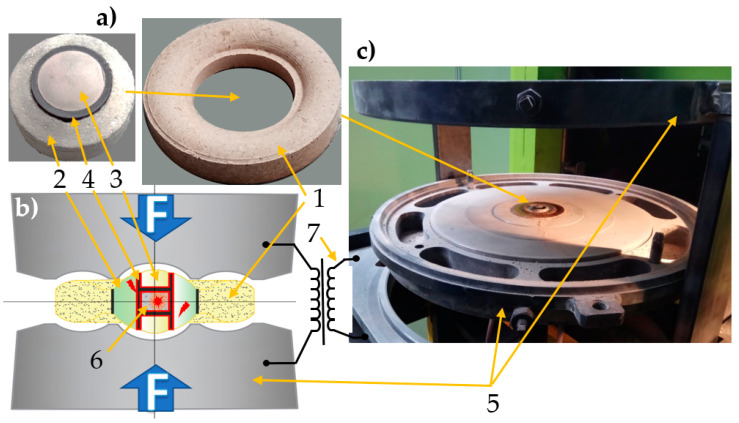
High Pressure HPHT/SPS Process: (**a**) view of the container: ceramic gasket 1—outer part, 2—inner central part, 3—ceramic disk, 4—graphite resistance heater, (**b**) cross—section of the container and the method of fixing the container between the anvils of the device, (**c**) view of the anvils and the container after sintering the sample—5—anvils, 6—the powder mixture placed in a toroidal chamber, 7—pulsed electrical generator.

**Figure 3 materials-16-01907-f003:**
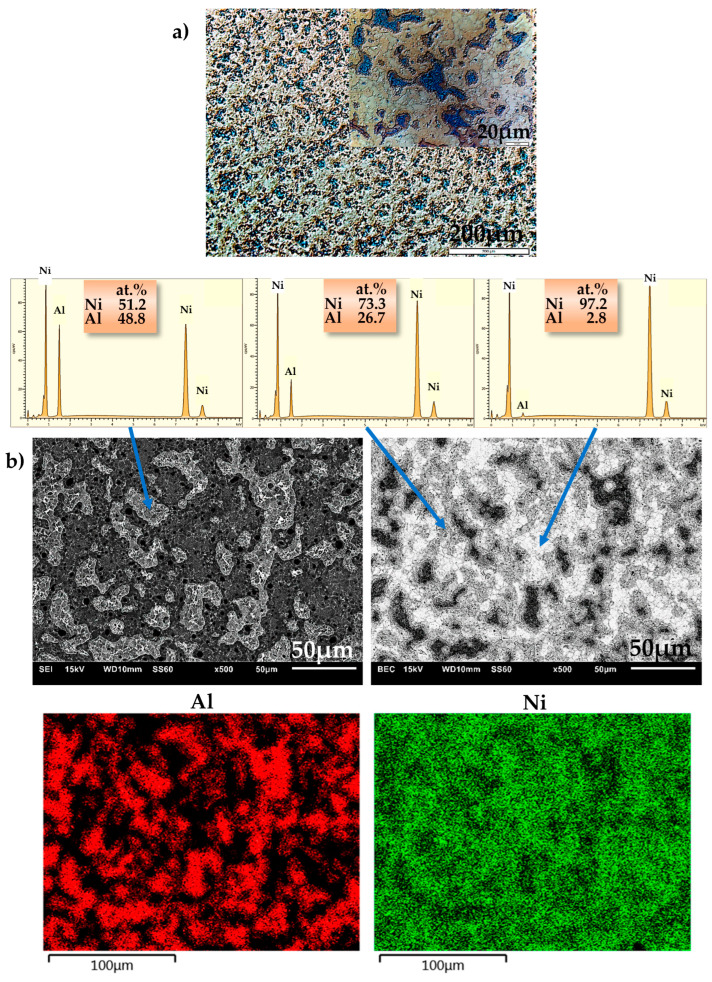
Microstructure of NiAl25 alloy: (**a**) optical microscope, Nomarski contrast; (**b**) SEI—secondary electron image—(**left**) and BEC—backscattered electron composition—(**right**) SEM micrograph, point-and-map analysis of the distribution of individual microstructural constituents.

**Figure 4 materials-16-01907-f004:**
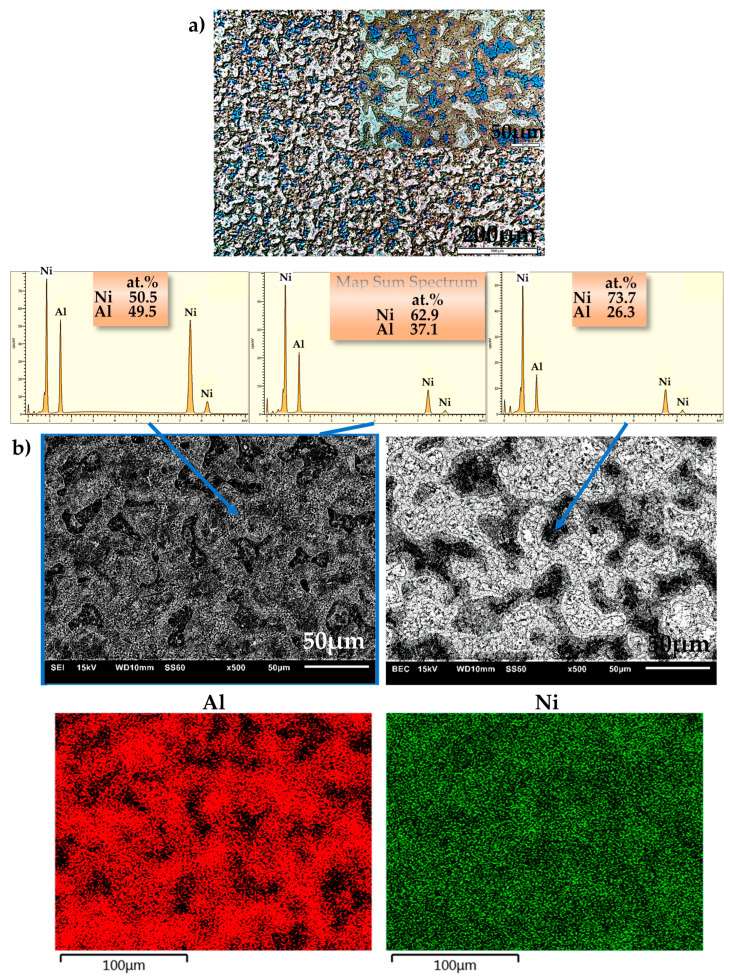
Microstructure of NiAl37 alloy: (**a**) optical microscope, Nomarski contrast; (**b**) SEI—secondary electron image—(**left**) and BEC—backscattered electron composition—(**right**) SEM micrograph, point-and-map analysis of the distribution of individual microstructural constituents.

**Figure 5 materials-16-01907-f005:**
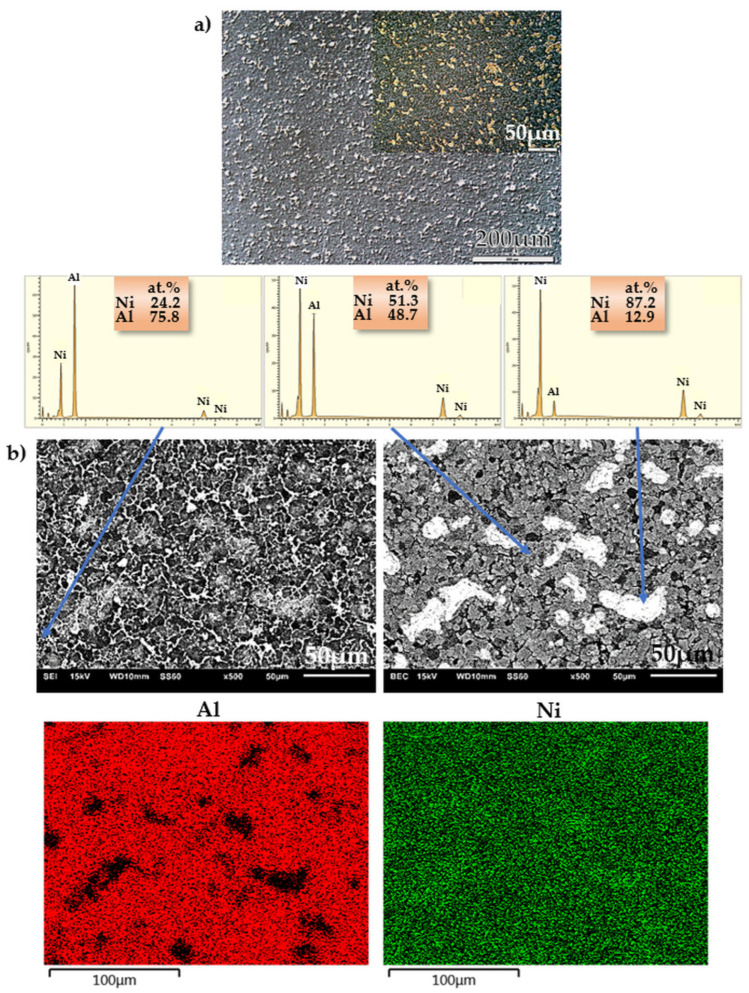
Microstructure of NiAl50 alloy: (**a**) optical microscope, Nomarski contrast; (**b**) SEI—secondary electron image—(**left**) and BEC—backscattered electron composition—(**right**) SEM micrograph, point-and-map analysis of the distribution of individual microstructural constituents.

**Figure 6 materials-16-01907-f006:**
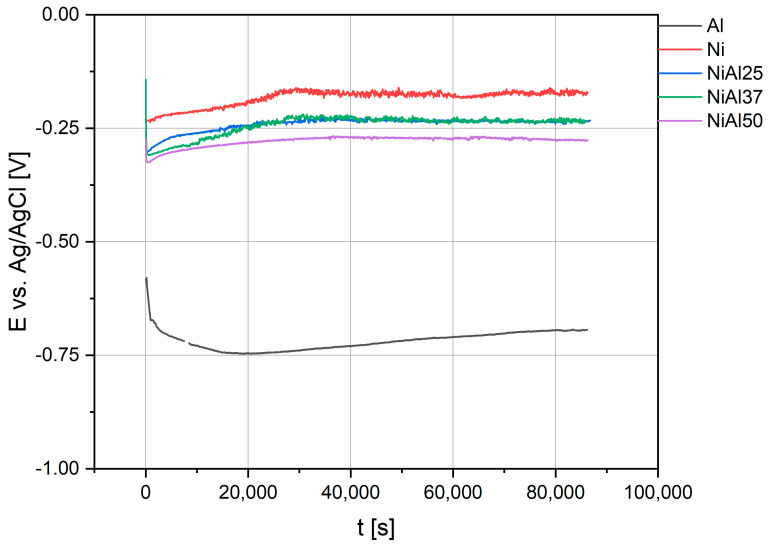
Open circuit potential of samples measured in 0.1 M H_2_SO_4_.

**Figure 7 materials-16-01907-f007:**
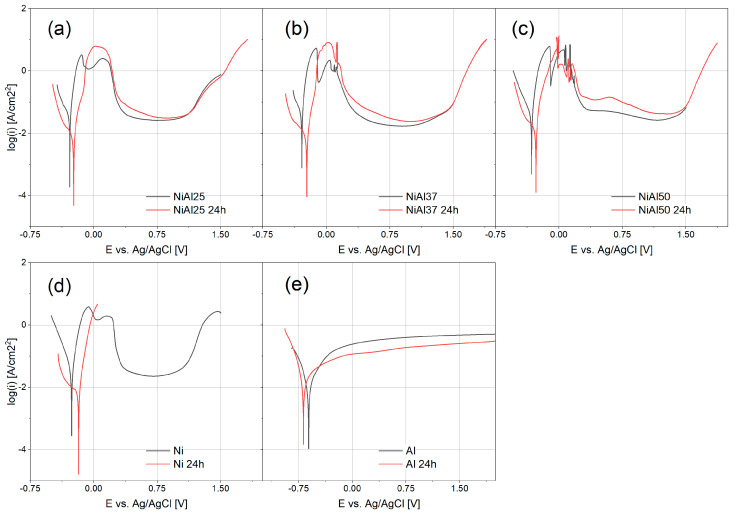
Polarization curves in 0.1 M H_2_SO_4_ for: (**a**) NiAl25, (**b**) NiAl37, (**c**) NiAl50, (**d**) Ni and (**e**) Al samples.

**Figure 8 materials-16-01907-f008:**
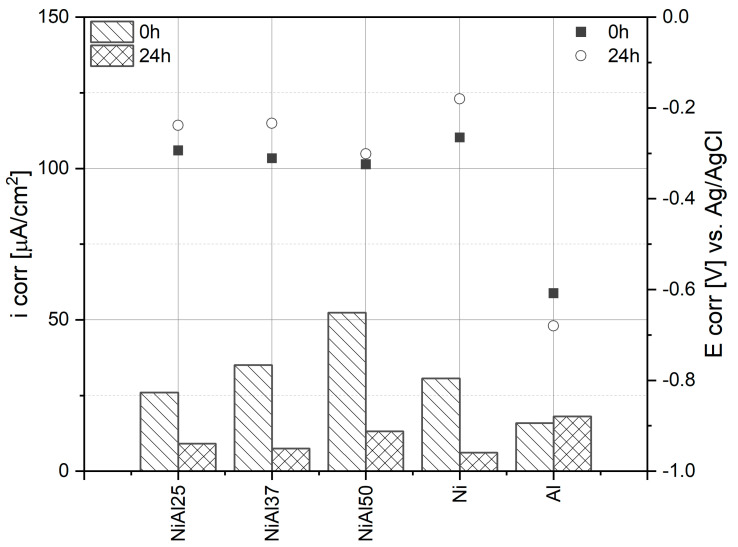
Corrosion current and corrosion potential of samples evaluated from polarization curves (Figure 7).

**Figure 9 materials-16-01907-f009:**
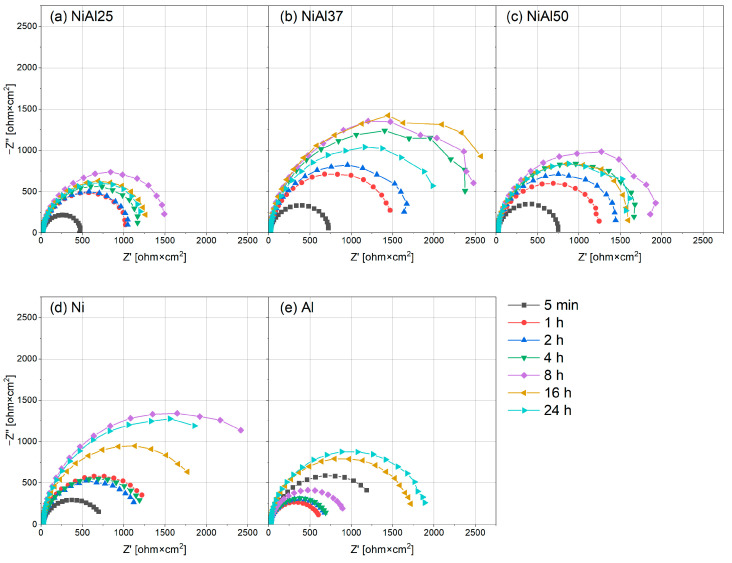
Nyquist representation of impedance data obtained in 0.1 M H_2_SO_4_, (**a**) NiAl25, (**b**) NiAl37, (**c**) NiAl50, (**d**) Ni and (**e**) Al.

**Figure 10 materials-16-01907-f010:**
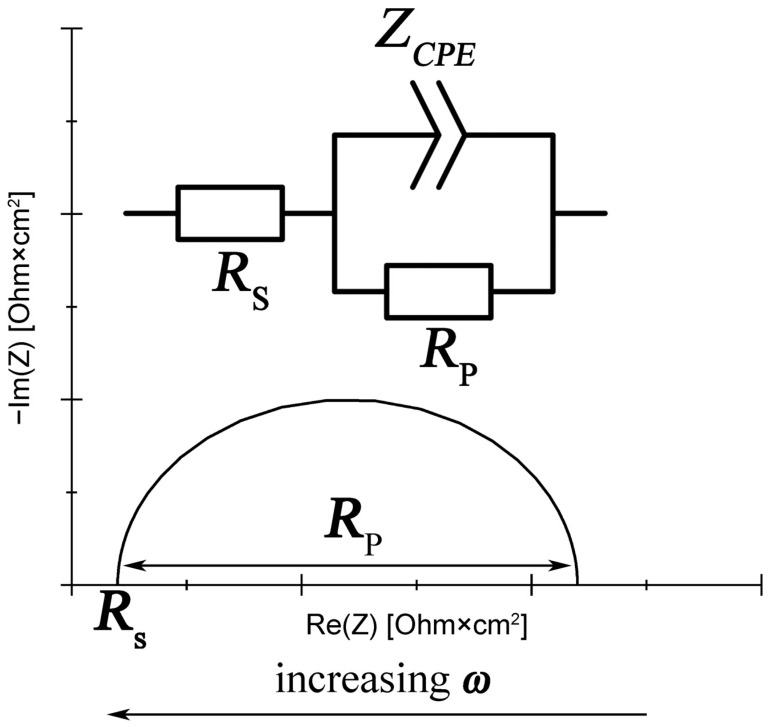
Equivalent circuit for the described corrosion system.

**Figure 11 materials-16-01907-f011:**
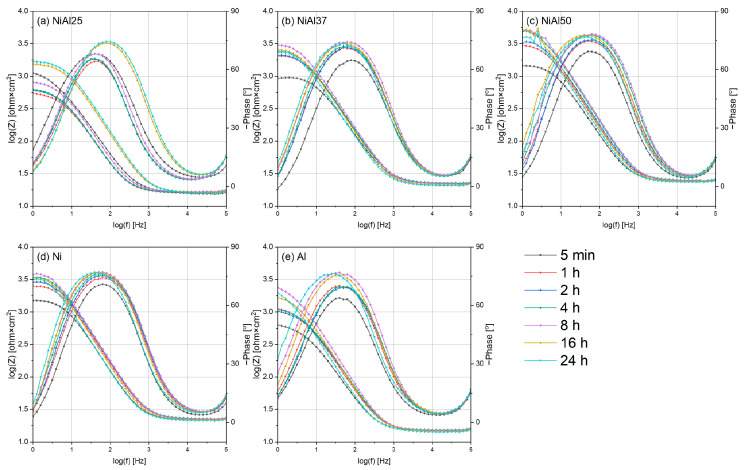
Bode representation of impedance data obtained in 0.1 M H_2_SO_4_, (**a**) NiAl25, (**b**) NiAl37, (**c**) NiAl50, (**d**) Ni and (**e**) Al.

**Table 1 materials-16-01907-t001:** The results of the density and HV10 hardness measurements.

Alloy Designation (According to at.% Al)	Density *ρ* (g/cm^3^)	Measurement Error (g/cm^3^)	Hardness HV10	Measurement Error HV10
NiAl25	7.38	± 0.02 (0.3%)	336	± 2.1 (0.6%)
NiAl37	6.67		399	± 5.5 (1.4%)
NiAl50	5.8		486	± 2.8 (0.6%)

**Table 2 materials-16-01907-t002:** Corrosion current, EW, density and corrosion rate obtained from polarization curves.

Sample	NiAl25		NiAl37		NiAl50		Al		Ni	
	0 h	24 h	0 h	24 h	0 h	24 h	0 h	24 h	0 h	24 h
*i_corr_* [uA/cm^2^]	25.97	9.10	35.05	7.52	52.40	13.15	15.92	18.11	30.64	6.14
*ρ* [g/cm^3^]	7.380		6.670		5.800		2.700		8.908	
*EW* [g]	22.56		19.74		17.14		8.99		29.35	
*CR* [mm per year]	0.260	0.091	0.339	0.073	0.506	0.127	0.173	0.197	0.330	0.066

**Table 3 materials-16-01907-t003:** Parameters obtained in fitting the EIS diagrams shown in Figure 9.

Sample	Time [min]	*R_s_* [Ω∙cm^2^]	*R_p_* [Ω∙cm^2^]_2_	C [µF∙cm ^−2^]	*n*	Chi-sqr × 10^−3^
Ni Al25	5	11.04	482	46.3	0.9108	3.03
	60	11.31	1074	36.9	0.9215	1.58
	120	11.32	1097	36.1	0.9240	1.68
	240	11.20	1238	35.1	0.9261	1.96
	480	10.96	1586	30.3	0.9363	1.23
	960	10.56	1362	37.1	0.9302	1.02
	1440	10.36	1330	44.6	0.9255	0.98
Ni Al37	5	12.46	737	40.2	0.9351	0.93
	60	12.65	1560	29.0	0.9437	0.62
	120	12.67	1764	25.3	0.9491	1.74
	240	12.53	2619	21.3	0.9545	1.34
	480	12.31	2762	19.7	0.9590	1.13
	960	12.08	3016	26.4	0.9480	1.03
	1440	11.85	2231	30.1	0.9463	1.03
Ni Al50	5	11.3	752	35.7	0.9518	1.79
	60	11.44	1296	28.1	0.9470	0.99
	120	11.52	1514	25.7	0.9513	1.82
	240	11.49	1791	24.8	0.9522	2.16
	480	11.36	2022	23.6	0.9594	3.61
	960	11.17	1713	27.3	0.9645	3.78
	1440	10.99	1743	33.7	0.9622	4.03
Ni	5	16.74	712	63.9	0.8976	0.45
	60	16.88	1339	41.7	0.9150	0.18
	120	16.89	1189	39.0	0.9150	0.26
	240	16.8	1273	37.7	0.9142	0.28
	480	16.73	2965	27.0	0.9452	0.49
	960	16.69	2065	32.4	0.9427	0.53
	1440	16.36	2777	44.4	0.9363	0.90
Al	5	17.66	1420	52.6	0.8784	0.74
	60	18.06	617	60.0	0.9152	0.31
	120	18.46	700	57.8	0.9199	0.27
	240	18.44	714	57.1	0.9208	0.28
	480	18.9	933	44.8	0.9277	0.25
	960	18.46	1756	16.0	0.9428	0.24
	1440	18.16	1926	14.3	0.9457	0.20

## Data Availability

Not applicable.

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
