# Peer review of "Corrosion Resistance of Nickel-Aluminum Sinters Produced by High-Pressure HPHT/SPS Method"

_materials, 2023, doi:10.3390/ma16051907_

Round 1
Reviewer 1 Report
1. The authors had better divide 2 and 3 paragraph into different sections (2.1, 2.2…) to make the article more logical. The order of conclusion should be 4.
2. XRD or SAED is needed to confirm the phase of NiAl or Ni3Al. The EDS result is not accurate.
3. The authors should give the theoretical densities of the three alloys and calculate the relative densities.
4. “Figure 8. Corrosion current and corrosion potential of samples evaluated from polarization curves (Fig. 9).” Fig.9 should be Fig.7.
5. The conclusion should be reorganized and the result of corrosion performance should be added.
Author Response
Please see the attachment
The manuscript was again reviewed by a native speaker.

Reviewer 2 Report
The manuscript presents an about the corrosion properties of nickel-aluminum sinters produced by HPHT/SPS method. However, the paper needs major revisions before it is processed further, some comments follow:
Abstract:
The abstract must be improved. Please highlight the importance and the novelty of this study. Also, introduce the methods used to characterize the material.
Introduction
Multiple citations have been introduced in bulk form "[1-8]", "[9-11]", "[20-24]" , "[31-33]" , "[36-40]" , "[46-49]" , "[51-53]" and not distributed in the text in accordance with the affirmations that must be supported. Please introduce citations in a specific position to ensure clear correspondence between the affirmations from the introduction section and the previous publication. Moreover, to avoid this type of citing, please cite the review type of studies.
Materials and methods
Line 105. Add the manufacturer of the powder.
Figure 1. If the authors want to add SEM micrographs, please introduce them in the Results section and discuss them.
Split this section into two subsections: one about materials and one about methods.
Line 186. What number means “N” types of samples?
Introduce the surface exposed.
Results and discussion
Split this section into many sections regarding the type of characterization.
Figure 3. The elemental flows are not readable. Please replace them. Also, in figures 4 and 5.
Figure 7 is not clear. Please replace it. Also, make them bigger. Also, add what represents each figure.
Please introduce a table with the values of current density, corrosion rate, corrosion potential and polarization resistance. And discuss all of them.
Figure 9. The figures are not clear. Please replace them. Also, rename the figure Nyquist plots of…. (a), (b), (c), etc.
Add one more figure containing the Bode plots.
Improve the section by comparing the results with other studies.
Conclusions
This section must be improved. Add also quantitative results, recommendations and suggestions.
Round 2
Reviewer 2 Report
The authors almost addressed all of my comments. However, are still some improvements to do:
As I recommended in the previous review. Please write in the title’s figure what represents figures (a), (b), (c) etc. That is available also in figure 9. I recommend introducing Bode diagrams in other figure and figure 9 to remain just for Nyquist diagrams. Also, I didn’t see some comments about them, so please add discussion regarding Bode diagrams.
Author Response
This has already been corrected in the text as recommended by the reviewer.